# Development of Wash-Durable Antimicrobial Cotton Fabrics by In Situ Green Synthesis of Silver Nanoparticles and Investigation of Their Antimicrobial Efficacy against Drug-Resistant Bacteria

**DOI:** 10.3390/antibiotics11070864

**Published:** 2022-06-27

**Authors:** Ashu Jain, Bhani Kongkham, Hariprasad Puttaswamy, Bhupendra Singh Butola, Hitendra Kumar Malik, Anushree Malik

**Affiliations:** 1Applied Microbiology Laboratory, Centre for Rural Development and Technology, Indian Institute of Technology Delhi, New Delhi 110016, India; ashujain8398@gmail.com; 2Environmental Biotechnology Lab, Centre for Rural Development and Technology, Indian Institute of Technology Delhi, New Delhi 110016, India; rdz188639@rdat.iitd.ac.in (B.K.); phari@rdat.iitd.ac.in (H.P.); 3Department of Textile and Fiber Engineering, Indian Institute of Technology Delhi, New Delhi 110016, India; bsbutola@textile.iitd.ac.in; 4Plasma Science and Technology Laboratory, Department of Physics, Indian Institute of Technology, New Delhi 110016, India; hkmalik@physics.iitd.ac.in

**Keywords:** silver nanoparticles, green synthesis, antimicrobial, wash-durable

## Abstract

An environment friendly and wash-durable silver nanoparticle treatment of cotton fabrics was carried out by in situ reduction of silver nitrate using *Azadirachta indica* leaf extract. The wash durability of the silver nanoparticles treatment on the cotton fabric was improved by pretreating the fabrics by mercerization and by adopting hydrothermal conditions of 120 °C temperature and 15 psi pressure for the in situ synthesis. The silver nanoparticle treated fabrics were characterized using scanning electron microscopy, colorimetric analysis and inductively coupled plasma mass spectroscopy. The coating of silver nanoparticles was seen to be dense and uniform in the scanning electron micrographs of the treated fabrics. An evaluation of the antibacterial efficacy of the silver nanoparticle treated fabric against antibiotic-resistant Gram-positive and Gram-negative strains was carried out. The antibacterial efficacy was found to be the highest against *Bacillus licheniformis*, showing 93.3% inhibition, whereas it was moderate against *Klebsiella pneumoniae* (20%) and *Escherichia coli* (10%). The transmittance data of a UV spectrophotometer (290–400nm) was used for measuring the UV protection factor of the silver nanoparticle treated fabrics. All the silver nanoparticle treated fabrics showed good antimicrobial and UV protection activity. The treatment was also seen to be durable against repeated laundering. This paper contributes the first report on a novel green synthesis approach integrating mercerization of cotton fabrics and in situ synthesis of nanoparticles under hydrothermal conditions using *Azadirachta indica* leaf extract for improved wash durability of the multifunctional fabric.

## 1. Introduction

There has been a boom in demand for antimicrobial textiles in recent times, created by higher standards of hygiene expected by the consumers [1]. Antimicrobial fabrics have been commercially available for decades; however, the emergence of super-microbes that are resistant to the antibiotics has given a strong impetus to research on treatment of fabrics with nanoparticles possessing broad-spectrum antimicrobial activity that is effective even against the antibiotic-resistant bacteria [2]. The antibiotics’ antimicrobial efficacy is due to their ability to hinder cell wall function, DNA replication and the expression of essential proteins. The bacteria have developed resistance to antibiotics by altering the target of the antibiotics, making them ineffective [3]. Antibiotic resistance in bacteria has been listed as one of the major threats to global health by the World Health Organization [4]. Antimicrobial nanoparticles achieve antimicrobial activity by adopting multiple simultaneous bactericidal pathways, making it difficult for bacteria to develop resistance [5]. Apart from combating bacteria on their own, nanoparticles can also be used for delivery of antibiotics and natural antimicrobial compounds [6]. Nanoparticles are therefore being widely studied as an alternative to antibiotics [7]. Amongst the antimicrobial nanoparticles, silver nanoparticles (AgNP) are the most preferred choice. This is because of their easy synthesis and immobilization on textiles [8] and because of them being a potent antimicrobial even at very low concentrations [9]. In fact, the concentration required for silver nanoparticles to be potent antimicrobials is much lower than the permissible safe level for humans [10]. The antimicrobial activity of silver nanoparticles is due to the constant release of Ag^+^ ions that cause disruption of the bacterial membrane and electron transport. The Ag^+^ ions also cause damage to the DNA [11]. AgNP-treated fabrics also offer protection against UV radiations as silver nanoparticles show absorbance in the UV-wavelengths. [12]. 

The most commonly adopted method for AgNP synthesis for textile treatment is by reduction of a silver salt using a reducing agent [8]. Currently, there is much emphasis on the use of green synthesized antimicrobial silver nanoparticles for textile modification to avoid harmful effects from use of toxic chemicals [13,14]. The green synthesis of silver nanoparticles is carried out by using a bio-extract as a reducing/stabilizing agent [15]. AgNP finishing to fabrics is given either by treating fabrics with pre-synthesized AgNP or by in situ synthesis of AgNP on fabrics [13]. Immobilization of pre-synthesized AgNP on fabrics was carried out by prolonged dipping of fabric in AgNP colloidal solution or by use of a binding agent for adhesion of AgNP to fabrics [16,17]. The durability of AgNP coating on fabrics treated with pre-synthesized AgNP was very low in absence of a binding agent. However, the use of graft polymers as a binding agent not only involved the use of additional chemicals but also resulted in nonuniform deposition of AgNP on fabrics and a reduction in antimicrobial activity of the AgNP [13,18]. In situ synthesis of AgNP on fabrics by both chemical [19], as well as green routes [20], resulted in a more uniform deposition of AgNP and an increase in durability of treatment but not sufficient enough to eliminate loss of AgNP on washing [13,19]. The use of AgNP-treated fabrics is increasing and so is the growing concern about the negative impact of AgNP leached during washing/usage on the aquatic life [21]. The durability of AgNP treatment to fabrics is therefore a prime concern. Alkali pretreatment to fabrics [22] and higher temperature synthesis [23] were seen to enhance the washing fastness of AgNP treatment to a certain extent. Mercerization is the alkali treatment of cotton fabrics, which is a very commonly adopted process in the textile industry for increasing the dyeability of the fabrics by activation of the cellulose, which is the main constituent of cotton [24]. Cellulose is a biopolymer having glucose as a building block. In natural condition, it has a convoluted structure, which swells and smoothens out during mercerization [24], causing crystalline disruption due to rearrangement of compact hydrogen bonds in the crystalline regions [25].

Plant extracts are the most preferred bio-extracts used for green synthesis of AgNP because of their easy availability, ease of handling and fast synthesis as compared to other bio-extracts [15]. Green AgNP synthesized using leaf extracts have been used for giving antimicrobial finishing to textiles [26]. The capping compounds in the bio-extracts are known to surface functionalize the AgNP, and if these compounds possess antimicrobial properties, the antimicrobial efficacy of the AgNP becomes enhanced. *Azadirachta indica* (neem) is a medicinal plant that belongs to the *Meliaceae* family and has well-established antimicrobial properties [27]. The antibiotic properties of the *Azadirachta indica* leaf extract are due to its constituent organic compounds such as nimbidin, nimbolide, mahmoodin, margolone, margolonone, and isomargolonon [28]. Herbal nanoparticles prepared by extensive ball-milling of *Azadirachta indica* dried leaves powder have also been used to give antimicrobial finish to cotton fabric [29]. However, moderate durability of the treatment was a limitation as the antimicrobial activity was found to diminish on washing. The leaf extract of *Azadirachta indica* is a storehouse of many phytochemicals [30], some of which act as reducing agents for AgNO_3_, leading to the synthesis of AgNP. Flavonoids and terpenoids were reported to act as reducing phytochemicals in *Azadirachta indica* leaf extract for AgNP synthesis [31]. The antimicrobial activity of AgNP synthesized using *Azadirachta indica* leaf extract has been widely reported [31,32,33,34]. However, *Azadirachta indica* leaf extract-mediated in situ synthesis of AgNP on fabric under hydrothermal conditions with suitable pretreatments for giving antibacterial finish to textiles has not been reported.

In the current study, a novel green synthesis approach was adopted for the development of multifunctional cotton fabrics having wash-durable antimicrobial and UV protective properties.

## 2. Results and Discussion

### 2.1. In Situ Synthesis of AgNP on Fabric under Different Conditions and Characterization

The use of *Azadirachta indica* leaf extract for efficient synthesis of AgNP has been reported by many studies where flavonoids and terpenoids were reported to act as reducing phytochemicals [31]. The in situ synthesis of AgNP on cotton fabric using *Azadirachta indica* leaf extract involves the use of reducing/ stabilizing agents from two sources, i.e., the phytochemicals, as well as cellulose, from the fabric [35]. Cellulose alone has also been used to reduce and stabilize silver to form AgNP [25,35]. In situ synthesis of AgNP is therefore expected to result in greater deposition of AgNP with increase in durability of AgNP treatment. This is because, unlike the case of treatment with pre-formed colloidal AgNP, in situ synthesis allows the growth of AgNP within the fibroids of cellulose, resulting in a higher entanglement of nanoparticles [36,37].

In this work, in situ synthesis of AgNP was carried out on mercerized, as well as un-mercerized, fabrics under both hydrothermal, as well as room temperature, conditions. The results pertaining to the AgNP content on the fabric, K/S values and grey scale rating for evaluating the wash durability of the AgNP treatment are shown in Table 1.

The different synthesis conditions influenced the concentration of silver in the AgNP-treated fabrics, which was seen by difference in brown coloration of the fabrics depicted by K/S values of various samples [14,38]. The K/S values can give a rough idea about the AgNP concentration on the fabric. K and S are the absorption coefficient and the scattering coefficient, respectively. For samples treated at room temperature, the K/S value for un-mercerized fabric was 1.04, whereas for samples pretreated with mercerization, it was 1.56. The highest K/S value of 1.72 was obtained when the AgNP treatment was carried out under hydrothermal conditions on mercerized fabric. Pretreatment by mercerization led an increase in the AgNP deposition on the fabrics. This was corroborated by the corresponding silver content of 51.69 µg g^−1^ fabric for the un-mercerized fabric as compared to 77.42 µg g^−1^ for the mercerized fabric. Mercerization allows infiltration of more silver atoms and formation of AgNP inside the cellulose matrix [22,39]. Further growth of these nanoparticles could cause their entrapment within the cellulose matrix because of the small pore size [25,39]. It has earlier been reported that the K/S values of AgNP-treated fabrics can be correlated with the Ag content of the AgNP-treated fabrics [14,36,40,41].

The gray scale comparative evaluation of washing durability of AgNP treatment on cotton fabrics under different experimental conditions shows a remarkable enhancement under hydrothermal conditions (Table 1). Next, the impact of AgNP in situ synthesis on fabrics under hydrothermal conditions was also evaluated on the resulting AgNP deposition on the fabrics. Synthesis at hydrothermal conditions was seen to cause an increase in AgNP deposition for both mercerized and un-mercerized fabrics. Hydrothermal synthesis of nanoparticles has been reported for faster synthesis of uniform sized nanoparticles, since the diffusion rate of atoms/ molecules increases under high temperature conditions [23]. In situ synthesis of TiO_2_ nanoparticles on cotton fabrics under hydrothermal conditions [18] was seen to increase the durability of the treatment. The same approach has been adopted in the present study for increasing the washing fastness of AgNP treatment on cotton fabrics. 

In the present study, the highest AgNP deposition on fabrics (Ag concentration of 85.46 µg g^−1^) was achieved on mercerized fabrics under hydrothermal conditions. 

The Scanning Electron Microscope images of fabric samples treated under the above conditions show homogenously distributed deposition of AgNP (Figure 1B,C) as compared to untreated fabric (Figure 1A).

The coloration of fabrics treated under hydrothermal conditions was also more uniform as compared to the fabrics on which in situ synthesis was carried out at room temperature. The higher deposition of AgNP on mercerized fabrics under hydrothermal conditions can be explained on the basis of mechanism of AgNP deposition on fabrics by in situ synthesis. Under wet conditions, the cotton fabric gets a slight negative charge [13,42]. Because of their negative surface charge, the positively charged Ag^+^ ions in solution get adsorbed on the textile fibers. This leads to the binding of Ag^+^ to the fibers through electrostatic interactions or van der Waals forces, where they are subsequently reduced to Ag atoms. These atoms subsequently form the AgNP [13] that get instantly immobilized on cotton fabric as templates. 

The morphological, topological properties and the chemical structure of the fibers of the fabric influence the amount of Ag adsorption [43]. Mercerization causes the fiber hair to swell, resulting in breaking of many hydrogen bonds in cellulose, causing a decrease in the crystallinity of cotton fibers [24]. Most textile fibres have two phases—crystalline (more ordered) and amorphous (less ordered, more accessible). While the crystalline part is responsible for stability, strength and other attributes, the amorphous part contributes to absorption reactivity etc. In crystalline part, only the –OH groups present on the surface are available for reaction/interactions with other moieties. Since the mercerized cotton has more amorphous content than un-mercerized cotton, it allows mercerized cotton to make available a greater number of –OH groups for reaction with other moieties. This increases the availability of the –OH group of cellulose, the prime sites of adhesion of Ag^+^, thus increasing the physical sorption of Ag in the fabric [43]. The average zeta potential of mercerized cotton fabric was −60.9 mV, which was much higher than the average zeta potential of fabric without mercerization which was −35.27 mV (Appendix A). Mercerization has also been reported to cause cotton fibers to swell and cause an increase in the pore size of fibers [39,44]. The combined effect of increased amorphousness, increased pore size due to mercerization and the increased diffusion rate of Ag^+^ in hydrothermal conditions must have caused increased deposition of AgNP on mercerized cotton fabrics by in situ synthesis of AgNP under hydrothermal conditions. The FTIR spectra of mercerized fabric before and after AgNP deposition were recorded (Appendix A), which depicted no change in the functional group after treatment, thereby ruling out any chemical bonding between fabric and AgNPs.

Figure 2 shows clear change in the post 20 washes coloration of cotton fabric where in situ synthesis of AgNP was done at room temperature without mercerization pretreatment. On the other hand, there was negligible loss of AgNP on washing when AgNP treatment was carried out under hydrothermal conditions on mercerized fabric, as can be seen by visual observation (Figure 2) and SEM micrographs (Figure 3). Hence, the conditions optimized in this study results in higher and uniform deposition of AgNP on the fabric, as well as makes the AgNP treatment wash-resistant.

### 2.2. Antimicrobial Activity of AgNP-Treated Fabrics on Multi-Drug Resistant (MDR) Strains

The susceptibility/resistance patterns of all the bacterial strains (*Klebsiella pneumoniae*, *Escherichia coli* and *Bacillus licheniformis*) were analyzed using the antibiotic disc diffusion assay (Appendix A). These bacterial strains were earlier reported as MDR strains and cause several infections with a high antibiotic resistance and infectivity rate in humans specially in hospital settings [45,46].

Different classes of antibiotics (beta-lactams, macrolide, fluoroquinolones, aminoglycoside, aminocoumarin, etc.) were used for this study. Strains *K. pneumoniae* and *E. coli* showed resistance to maximum number of antibiotics compared to *B. licheniformis*. However, all the strains showed resistance to a minimum of eight antibiotics of different classes. Both *K. pneumoniae* and *E. coli* showed comparatively greater resistance towards the third and fourth generations antibiotics such as imipenem, meropenem, levofloxacin, sparfloxacin, etc, whereas *B. licheniformis* was found susceptible for the above-mentioned third and fourth generations antibiotics (Figure 4). Among the three strains, *B. licheniformis* showed significant (*p* < 0.05) biofilm formation, followed by *K. pneumoniae* (Figure 5).

The disc diffusion assay was also performed for all the strains using both the AgNP-treated and untreated fabric. No zone of inhibition was seen for *K. pneumoniae* and *E. coli*, while a small zone of <1 mm was seen around the treated fabric in the case of *B. licheniformis* (Appendix A). A possible explanation for such observation is that either these strains are resistant against the treated fabric or there is a lack of diffusion of the coated AgNP through the nutrient agar. 

In the broth assay, AgNP was found leaching into the media from fabrics. Further, *B. licheniformis* was found to be the most susceptible strain showing 93.3% inhibition followed by *K. pneumoniae* (20%) and *E. coli* (10%) (Figure 6A). A similar trend was observed in both the FDA (Fluorescent diacetate) assay (Figure 6B) and spread plate method (Figure 6C and Appendix A). 

As evident from the results both bacterial growth and microbial enzymatic activity were not significantly inhibited in case of *K. pneumoniae* and *E. coli* in comparison to *B. licheniformis*. Though all the strains are characterized as MDR, they showed differential susceptibility/resistance pattern towards AgNP-treated fabric. Earlier studies also showed such differential susceptibility/resistance pattern while using AgNP and AgNP-coated materials as antimicrobial agents against bacterial strains such as *Staphylococcus aureus*, *Staphylococcus epidermidis*, *Pseudomonas* spp., etc. [47,48,49,50]. Further, quantification of the fabric leached AgNP into the broth, revealed a significant difference among the tested bacteria. In the case of *K. pneumoniae* and *E. coli*, the concentration of AgNP in broth was found to be 6953.43 ppb and 8186.37 ppb which was significantly (*p* < 0.05) lower than the control (10216.88 ppb) and *B. licheniformis* (9999.47 ppb) (Figure 7).

This data supports the nature of AgNP resistance in *E. coli* and *K. pneumoniae*. This could be due to the aggregation or precipitation of AgNP in the liquid media by the bacterial strains as reported earlier. When bacteria were exposed to the sub-inhibitory concentration of AgNP repeatedly, they developed resistance by the production of an adhesive protein flagellin which aggregates the AgNP thereby reducing their antibacterial effect [51]. Moreover, bacterial resistance to AgNP can be intrinsic (efflux pump, porin downregulation, etc.) or extrinsic (mutation, plasmid containing resistance gene, etc.) [51]. The results obtained in our study also indicated that the resistance recorded by *K. pneumoniae* and *E. coli* may be due to such activity, which was evidenced by lower concentration of AgNP in culture filtrate.

Earlier studies also evidenced the antimicrobial and antibiofilm activity of AgNP coated material, and their results also support our study [49,52]. Moreover, from the studies it can be concluded that the bacterial growth in response to AgNP is concentration dependent, with the minimum inhibitory concentration (MIC) ranging from 11.25 to 45 µg mL^−1^ [49] and AgNP the concentration-dependent prolongation of bacterial growth lag phase [52]. Additionally, AgNP showed complete biofilm formation inhibition of methicillin-resistant *Staphylococcus aureus* and *S. epidermidis* at a low concentration of 50 µg mL^−1^ [49]. In our study, *B. licheniformis* was inhibited by AgNP in broth assay. Additionally, SEM-EDAX shows *B. licheniformis* colonization on the surface of the untreated fabric, while there was no such colonization in case of the treated fabric (Figure 8). The probable mechanism of AgNP antibacterial and antibiofilm activity can be AgNP induced higher reactive oxygen species production inside the cell and arresting of bacterial metabolic pathways leading to reduction in biofilm production [53,54]. However, the precise mechanism of AgNP action remain uncertain. Further, in case of *E. coli* and *K. pneumoniae* resistance was seen against AgNP in broth assay. Previous studies reported the resistance developed against AgNP in clinically important bacterial isolates [51,55] and such results are in congruence with our experimental results. Therefore, though the use of AgNP as antimicrobial and antibiofilm agent in hospital and other industrial set ups is promising, such usage should be done with proper caution and research.

### 2.3. Effect of AgNP Content on UV- Blocking Efficiency of AgNP-treated Fabrics

Silver nanoparticles have UV-protective properties [12], because of which, the AgNP-treated fabrics were seen to have much enhanced UV blocking properties as compared to pristine fabric. The UPF values of untreated fabrics and AgNP-treated fabrics with different silver contents are shown in Table 2. 

All the AgNP-treated fabrics had a UPF ranging from 33.4–89.9. According to standards (Australia/New Zealand, AS/NZS 4399), fabrics with a UPF value of 25–40 provide ‘very good’ protection against UV radiations while fabrics with a UPF > 40 provide ‘excellent’ protection against UV radiations.

## 3. Materials and Methods

### 3.1. Materials and Chemicals

Cotton spandex blend fabric (Desized, scoured and bleached) was procured from Vardhman Textiles limited. The warp and weft count of the fabric were 50 each. The warp was 100% cotton and the weft of the fabric was cotton spandex blend. Ends per inch of the fabric were 158 and picks per inch of the fabric were 80. Silver nitrate and sodium hydroxide used in the study were of analytical grade and were procured from Merck, India. Ultrapure water (Organo Biotech laboratories private limited, New Delhi, India), Agar-Agar, Nutrient Agar and other chemicals were of analytic grade and used without further purification. Fresh leaves of *Azadirachta indica* were collected from Indian Institute of Technology Delhi, New Delhi, India. Non-ionic detergent was used for assessing the wash durability of silver nanoparticle treatment on fabrics. 

Bacterial cultures *Klebsiella pneumoniae* MCC 2451 and *Escherichia coli* MCC 3671 were procured from the culture deposit of National Center of Cell Science (NCCS), Pune. These strains were earlier reported to be multi-drug resistant (MDR) strains [56,57]. Additionally, the bacterial culture *Bacillus licheniformis* strain CRDT-EB-3.1 which was originally isolated from green leafy vegetables and characterized for their antibiotic resistance was also used [58]. It was obtained from the bacterial culture collection of Environmental Biotechnology Lab, IIT Delhi. The strains were sub-cultured regularly on nutrient agar plates and for long term storage glycerol stock (40%) were prepared and maintained at −80 °C.

### 3.2. Procedure

The aim of this work was to optimize the in situ green synthesis of AgNP on cotton fabric for uniform deposition and wash durability. Hence, in situ synthesis of AgNP was carried out on mercerized, as well as un-mercerized fabrics under both hydrothermal, as well as room temperature, conditions. The experimental protocol involved the preparation of the leaf extract, a mercerization pretreatment of cotton fabrics, followed by the in situ synthesis. 

#### 3.2.1. Preparation of Leaf Extract

A 20% *w/v* leaf extract was prepared by taking 40 g of washed and shade-dried fresh leaves of *Azadirachta indica*. The leaves were finely chopped and heated in 200-mL deionized water at 65–70 °C for 10 min. On cooling, the extract was filtered using Whatman grade 1 filter paper and stored at 4 °C for subsequent use. The extract was used within 10 days of its preparation.

#### 3.2.2. In Situ Synthesis of AgNP on Cotton Fabric at Room Temperatures

In situ synthesis of AgNP was carried out on cotton fabrics by dipping fabric samples of 10 cm × 5 cm in a reaction solution containing 2mM AgNO_3_ and 20% (*w/v*) leaf extract in 9:1 ratio for 10 min at room temperature conditions (26 ± 2 °C) and pH 7.1. The material to liquor (ML) ratio, which is the ratio of the volume of the reaction mix to the weight of the fabric (mL g^−1^), was kept at 40. 

#### 3.2.3. Mercerization Pretreatment and In Situ Synthesis of AgNP on Cotton Fabric at Room Temperatures 

With an aim to improve the in situ synthesis of AgNP, mercerization of cotton fabrics was done as pretreatment. Fabric pieces were dipped in 2.5 M NaOH for 1 min with gentle stirring and then washed with running water. The fabric pieces were then dipped in 1% acetic acid for neutralization and washed again with running water. In situ synthesis of AgNP on this mercerized cotton fabric was done at room temperatures under conditions described in Section 3.2.2.

#### 3.2.4. In Situ Synthesis of AgNP on Cotton Fabric under Hydrothermal Conditions 

Both the mercerized, as well as un-mercerized, fabric samples were subjected to in situ synthesis of AgNP under hydrothermal conditions (temperature ≥ 120 °C and pressure of 15 psi for 180 min) in laboratory IR dyeing machine (Daelim Starlet Lab Infrared dyeing machine). The rest of the reaction conditions (2mM AgNO_3_ and 20% *w/v* leaf extract in 9:1 ratio, pH 7.1 and ML ratio 40 mL g^−1^) were kept the same. 

The fabric samples from each of the above treatment were analyzed for color measurements, silver content, surface morphology and surface groups, wash durability, UV protection and antibacterial activity, as described in subsequent sections. 

### 3.3. Measurements

#### 3.3.1. Color Measurements

The brown coloration of the samples occurs due to the unique optical properties of the silver nanoparticles and varies depending on the size, shape and the concentration of the silver nanoparticles [59] deposited on the cotton fabric samples. The samples were subjected to colorimetric-analysis by employing a spectrophotometer having pulsed xenon lamps as light source (Gretag Macbeth Color Eye 7000A Martinsried, Germany), using D65 illuminant with 10^0^ observer, d/2 viewing geometry and measurement area of 2 mm. The strength of the color of the fabric is related to its absorption property and the color strength is frequently computed as ratio of K/S values. The relation between the absorbance and reflectance is given by Equation (1) in view of ‘Kulbela-Munk’ theory
(1)$$\frac{K}{S}=\frac{{\left(1-R\right)}^{2}}{2R}$$
where *R*, K and *S* are the reflectance, absorption coefficient and the scattering coefficient, respectively. This equation is relevant for samples having opacity of more than 75% and is very helpful in formulating colors for textile-industry. For textiles, it is assumed that the scattering *(S*) depends on the properties of the substrate (fabric), whereas the absorption (K) depends on the properties of the colorant. The K/*S* values can also give a rough idea about the AgNP concentration on the fabric, since the Kulbela-Monk equation is roughly linear with respect to the colorant concentration [56]. Measurements were taken at wavelengths 360–750 nm. 

#### 3.3.2. Detection of Silver Content

A small piece of treated fabric (~3–4 mg) was weighed and digested in 5 mL HNO_3_ using a microwave digestor (Anton Parr, Graz, Austria). The digested sample was centrifugated, and ICP-MS (Inductively Coupled Plasma Mass Spectrometry) analysis was carried out on the supernatant to determine the concentration of silver on the tested fabric using Agilent 7000 ICP-MS. 

#### 3.3.3. Scanning Electron Microscopy

A 5 mm × 5 mm of various fabric samples treated with silver nanoparticles under different experimental conditions were mounted on the sample stub and sputter coated with Au. The silver nanoparticle deposition on the samples was then observed using scanning electron microscope (ZEISS EVO 50, Oberkochen, Germany). 

#### 3.3.4. Washing Process

The AgNP-treated fabrics were tested for color-fastness properties in accordance with the ISO standard method AATCC 61(2006). Each washing cycle as per the test standard is equivalent to five cycles of home laundering at an ambient condition of 38 ± 3 °C. We conducted tests equivalent to twenty home laundering.

#### 3.3.5. Determination of Washing Durability of AgNP Treatment

The washing durability is indicated by the extent of change in color of the fabric after washing since the brown coloration of the fabrics is caused by silver nanoparticles. The washing durability is graded using a grey scale comparison of fabric color pre- and post-washing; categorized from 1–5, where higher number indicates superior washing durability. The evaluation of washing durability is done as per test standard AATCC EP1, 2012. The Gray Scale enables visual evaluation of changes in textile coloration caused by tests for colorfastness. 

#### 3.3.6. Fourier-Transform Infrared Spectroscopy (FTIR)

FTIR spectroscopy (Perkin Elmer One spectrum) of untreated mercerized cotton fabric and mercerized fabric treated with AgNP were carried out to determine the presence of chemical bonding between the fabric and the AgNP. The FTIR analysis was performed with KBr pellets. The FTIR was recorded in a diffuse reflection mode at a resolution of 4 cm^−1^. 

#### 3.3.7. Zeta Potential (ζ Potential) Measurements

The zeta potential (Anton Paar SurPAAS 3) of mercerized cotton fabric and un-mercerized cotton fabric was carried out to determine the surface charge in dry conditions. The classic streaming potential and streaming current method was used for carrying out a direct analysis of the surface ζ potential.

#### 3.3.8. Ultraviolet Radiation Blocking EFFICACY

The UV-blocking efficacy of the AgNP-treated fabrics was evaluated by recording the transmission of UV radiations through the treated fabric with a UV transmittance analyzer, model UV 2000, Labsphere. The range of wavelength of the radiation measurement was 290–400 nm, with an interval of 5 nm. For each sample, an average of five measurements from different areas was considered. Using the transmission data, the UV radiation protection for the AgNP-treated fabrics is calculated according to AATCC Test method 183-2004 (AATCC Test Method 183-2004; Transmittance or Blocking of Erythemally Weighted Ultraviolet Radiation Through Fabrics; AATCC Technical Manual, (2010) Vol.85 p. 318–321). The ultraviolet radiation protection factor (UPF) was computed using Equation (2)
(2)$$\mathrm{UPF}=\frac{\sum_{290\mathrm{nm}}^{400\mathrm{nm}}E_{\lambda }\times S_{\lambda }\times \Delta \lambda }{\sum_{290\mathrm{nm}}^{400\mathrm{nm}}E_{\lambda }\times S_{\lambda }\times T_{\lambda }\times \Delta \lambda }$$

Here, *E_λ_*, *S_λ_*, *T_λ_* and ∆*λ* are the relative erythemal spectral effectiveness, the solar spectral irradiance, the average measured transmittance and the interval of measured wavelength (5 nm), respectively. The values of *E_λ_* and *S_λ_* were obtained by conducting tests as per AATCC Test method 183.

### 3.4. Antimicrobial Activity of AgNP-Treated Fabrics on Multi-Drug Resistant (MDR) Strains

The ability of the AgNP-treated fabric to inhibit three MDR bacteria namely *Klebsiella pneumoniae*, *Escherichia coli* and *Bacillus licheniformis* were tested using both microdilution assay and disc diffusion method. Further, the anti-biofilm activity of the treated fabric was test using the standard biofilm assay using crystal violet dye. 

#### 3.4.1. Biofilm Assay

Some (10 µL) overnight grown bacterial culture was added to 250 µL of nutrient broth in a 96-well plate and incubated at 35 ± 2 °C for 36 h on an incubator in stagnant condition. After incubation period, absorbance was read at 610 nm using a microtiter plate reader (M491-Epoch reader, Bio Tek Instruments, Inc., Winooski, VT, USA), and the broth was decanted out without disturbing biofilm adhered on the bottom surface of the plate. The plates were then washed thrice with phosphate buffer saline to remove loose or unadhered bacteria. The biofilm staining was performed following the standard procedure [60]. Some (125 µL) 0.1% crystal violet was added to the wells and incubated for 10 min at 35 ± 2 °C. The plate was then washed thrice with distilled water and dried. 250 µL of ethanol (95%) was added to the wells and incubated for 10 min. After incubation, the colored solution was aspirated to new 96 well microtiter plates and absorbance was read at 550 nm. The biofilm forming capabilities of each bacterium was performed in triplicates and repeated thrice.

#### 3.4.2. Disc Diffusion Assay

Antibiotic disc diffusion assay was performed using antibiotic octa disc (HiMedia OD291R, OD011R and OD258R). Briefly, a loopful of overnight grown bacterial cultures was streaked on nutrient agar plates uniformly. The antibiotic octa disc was placed at the center of bacteria-streaked nutrient agar plates. The plates were then incubated for 16–18 h at 35 ± 2 °C. After incubation period, the zone of inhibition appeared around the antibiotic disc were measured in mm. The experiments were conducted in triplicates and repeated three times for each bacterial isolate. For resistance/susceptibility of selected bacteria against silver nanoparticles coated fabric. The disc diffusion assay was performed using silver nanoparticles coated and uncoated fabrics (6 mm diameter) instead of standard antibiotic discs.

#### 3.4.3. Bacterial Inhibition in Broth Assay

One coated and uncoated fabric (6 mm diameter in size) was placed in a 96 well plate. Some (250 µL) nutrient broth was added on each well along with 10 µL of overnight grown bacterial broth. The plate was incubated at 35 ± 2 °C for 36 h. After incubation, 200 µL of the broth was withdrawn and placed in a new 96-well plate. The reading of the broth was then taken at 610 nm. Further, the broth was syringed filtered (Axiva 0.22 µm syringe filter), and the concentration of AgNP leached out from the fabric was detected using the ICPMS facility (ICPMS system, Agilent 7900) at Central Research Facility, IIT Delhi. 

In another set of same experiments, after incubation, 200 µL of the broth was taken and diluted with 1.8 mL of saline solution (0.85%), followed by subsequent serial dilutions until 10-5. For FDA (Fluorescent diacetate) assay, 500 µL of all dilutions and stock samples were taken in a sterile falcon tube and 25 mL of 60 mM sodium phosphate buffer (pH 7.6) was added. Further, 250 µL of 4.9 mM FDA lipase substrate solution (Sigma, St. Louis, MO, USA) was added, and the tubes were mixed thoroughly. The tubes were then incubated for 3 h at 37 ± 2 °C. After the end of incubation period, 2 mL of acetone was added to all the tubes and mixed properly. The mixture was centrifuged at 8000 rpm at 4 °C. The supernatant was passed through Whatman no.2 filter paper and the 200-µL filtrate was transferred to a microtiter plate and absorbance was measured at 490 nm [61]. 

Additionally, the same dilutions (10^−1^ to 10^−5^) were used to quantify bacterial colony forming unit (CFU) through spread plate method. Briefly, 50 µL of different dilutions of the samples was spread on nutrient agar plates using a glass spreader. The nutrient agar plates were then incubated for 16–18 h at 35 ± 2 °C. After the incubation period, the number of colonies formed in both control and test samples were counted using a colony counter. The number of colonies formed was represented as CFU mL^−1^ of the sample. 

For all the above-mentioned experiments, Control was the broth of the well where untreated fabrics were placed. Percentage bacterial inhibition of treated fabrics was calculated over control. The experiments were done in triplicates and repeated twice for all the three bacterial strains studied.

#### 3.4.4. Determination of Bacterial Biofilm Formation on Fabric

Fabric coated with silver nanoparticles (NP) and uncoated fabric were placed in 96 well plate. 250 µL of nutrient media was added to the wells followed by 10 µL of overnight grown bacterial broth (*Bacillus licheniformis*). The plate was incubated for 48 h at 35 ± 2 °C. After incubation, the fabrics were removed from the plate, transferred to a new 96 well plate and biofilm formation on fabric was tested through SEM-EDAX analysis. Briefly, the fabrics were washed with PBS thrice, followed by overnight fixation with 5% glutaraldehyde prepared in 0.1 M phosphate buffer (pH 7.2). After fixation, the fabrics were washed with 0.1 M phosphate buffer thrice. The fabrics were then dehydrated using ethanol of different concentrations (10%, 30%, 50%, 70%, 80%, 90% and 100%). The dehydrated samples were immersed in hexamethyldisilazane (HMDS) for 10 min. Then, the samples were put in a petri plate and kept in a desiccator overnight with the cover slightly open. Tabletop SEM/EDAX-element analysis/EDAX-mapping (Hitachi High Technology) of the samples were performed in duplicates at NRF (Nanoscale Research Facility) facility of Indian Institute of Technology Delhi, India.

#### 3.4.5. Statistical Analysis

The data obtained from the microbiology experiments were subjected to one-way analysis of variance (ANOVA) test. To compare the differences between the means, Tukey post-hoc test was performed at a significance level of *p* < 0.05 using the SPSS software, version 25.0 (SPSS Inc., Chicago, IL, USA). All the experiments were conducted in triplicates and the results are shown as mean ± standard error.

## 4. Conclusions

The silver nanoparticles treated fabrics were seen to be effective against some antibiotic resistant strains of bacteria indicating differential susceptibility/resistance of bacteria to silver nanoparticles. The adoption of novel strategy of carrying out green in situ synthesis of silver nanoparticles under hydrothermal conditions using *Azadirachta indica* leaf extract as a reducing/stabilizing agent on mercerized cotton fabric resulted in fabrics with higher deposition of AgNP, as well as an improved wash durability. The AgNP treatment is devoid of use of toxic chemicals during synthesis and superior washing fastness enables the AgNP treatment to be durable and ‘greener’ during use because of minimized nano-pollution by loss of nanoparticles in wash-water. Further insights into the mechanisms governing improved in situ green synthesis of AgNP on mercerized fabric under hydrothermal conditions are needed to enhance the understanding, as well as potential applications.

## Figures and Tables

**Figure 1 antibiotics-11-00864-f001:**
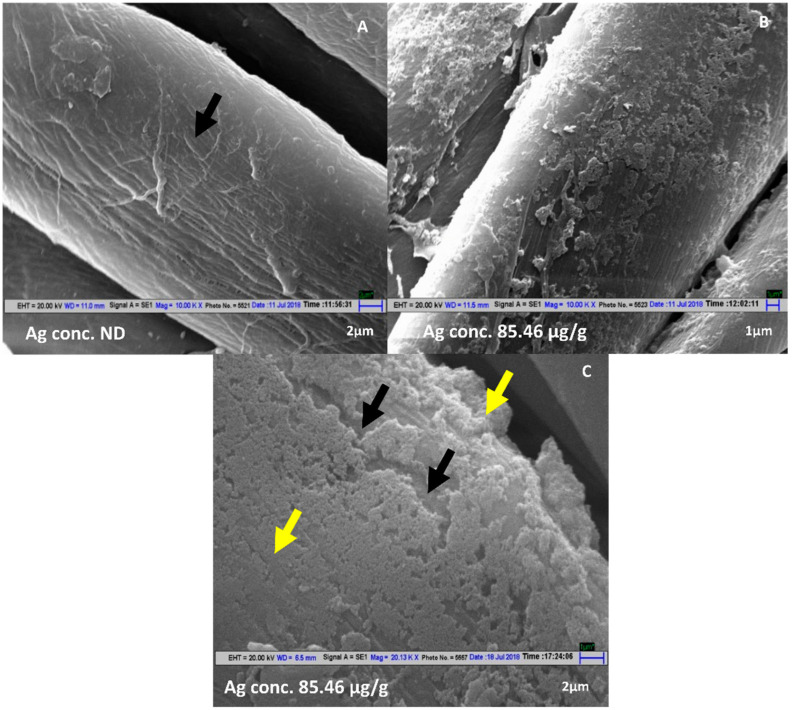
SEM micrographs of mercerized fabrics: (**A**) No AgNP treatment. (**B**) A 2 mM AgNO_3_, 20% Leaf Extract, Mixing ratio 9:1, pH 7.1, hydrothermal conditions (Magnification 10KX). (**C**) A 2 mM AgNO_3_, 20% Leaf Extract, Mixing ratio 9:1, pH 7.1, hydrothermal conditions (Magnification 20 KX). Black arrows indicate the cotton fiber and yellow arrows indicate AgNP deposition the on fiber. ND represents Not Detected.

**Figure 2 antibiotics-11-00864-f002:**
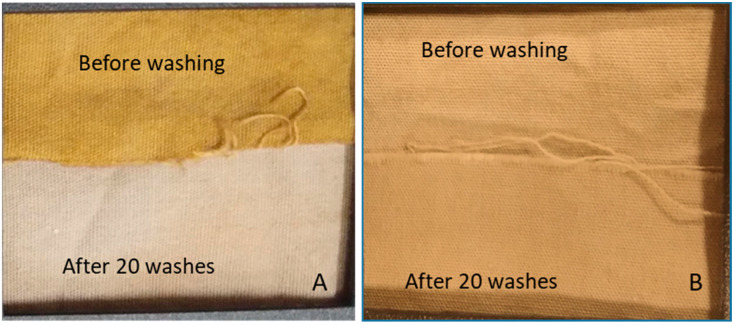
Washing durability of cotton fabric treated by AgNP by in situ synthesis by reacting 2 mM AgNO_3_ with 20% (*w/v*) *Azadirachta indica* leaf extract in a ratio of 9:1 at pH 7.1. (**A**) Un-mercerized sample. AgNP synthesis at room temperature. (**B**) Mercerized sample. AgNP synthesis at hydrothermal conditions.

**Figure 3 antibiotics-11-00864-f003:**
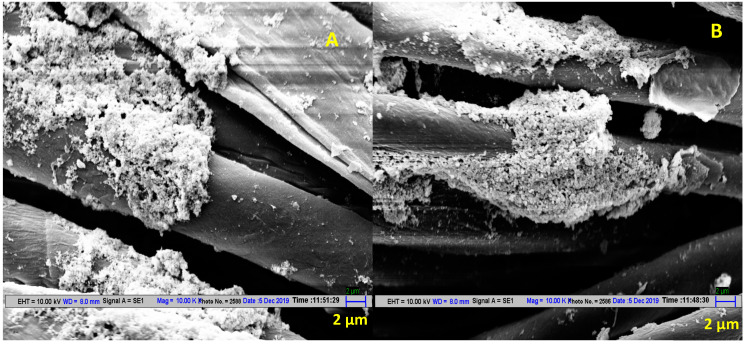
SEM micrographs of mercerized cotton fabric treated by AgNP via in situ synthesis by reacting 2 mM AgNO_3_ with 20% (*w/v*) *Azadirachta indica* leaf extract in a ratio of 9:1 at pH 7.1. (**A**) Before washing. (**B**) After 20 washes.

**Figure 4 antibiotics-11-00864-f004:**
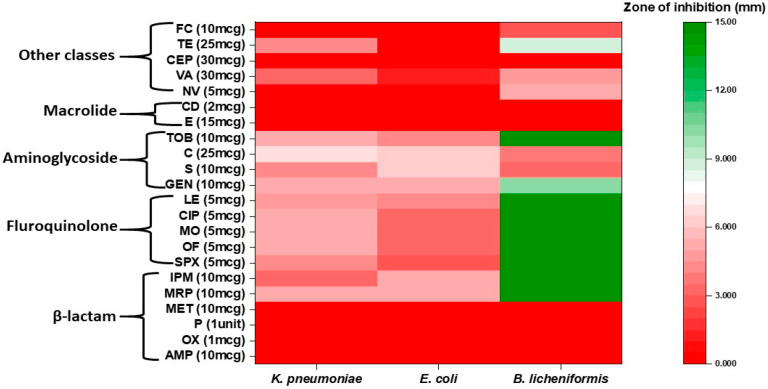
Heatmap representing antibiotic resistance/susceptibility pattern of all the three test bacteria.

**Figure 5 antibiotics-11-00864-f005:**
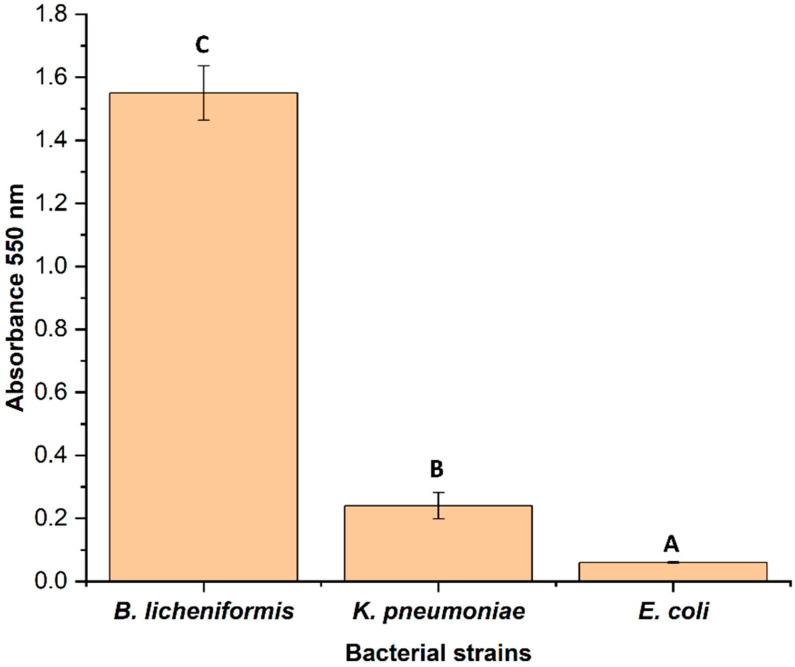
Biofilm forming capabilities of these test bacteria on the surface of wells of microtiter plate. Bars having different alphabet letters are statistically different.

**Figure 6 antibiotics-11-00864-f006:**
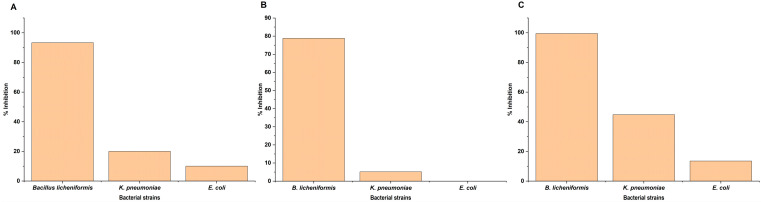
Broth assay indicating differential resistance/susceptibility of bacteria to the AgNP-treated fabric. (**A**) Bacterial growth inhibition by measuring optical density. (**B**) FDA bioassay. (**C**) Spread plate method.

**Figure 7 antibiotics-11-00864-f007:**
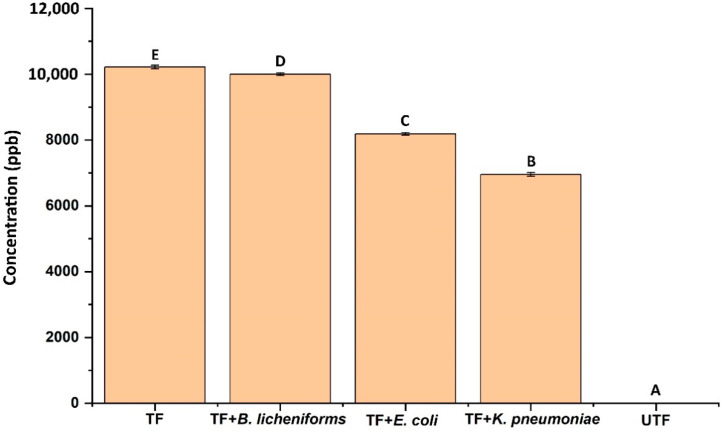
Concentration of AgNP detected in the culture broth treated with different test bacteria. TF denotes treated fabric and UTF denotes untreated fabric. Bars having different alphabet letters are statistically different.

**Figure 8 antibiotics-11-00864-f008:**
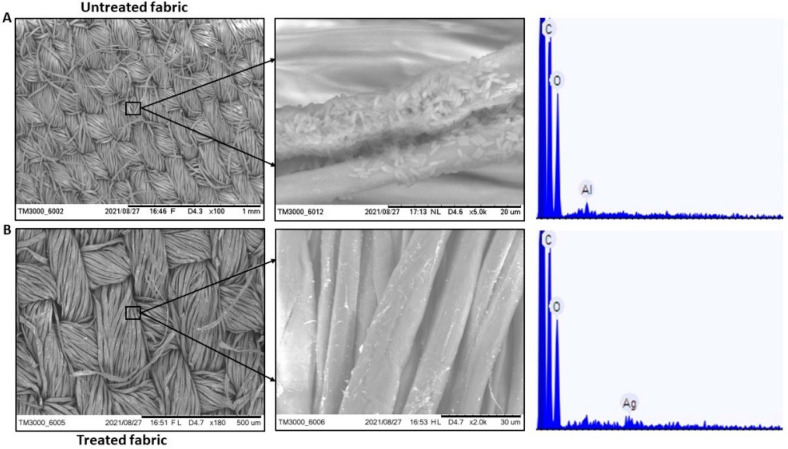
SEM-EDAX images of (**A**). Untreated fabric showing bacterial colonization and absence of Ag on the surface (**B**). Treated fabric showing no bacterial colonization and presence of AgNP on the surface. (EDAX value: 0.139 atomic%, acquisition time 60 s, accelerating voltage 15.0 kV, image width 316.0 µm).

**Table 1 antibiotics-11-00864-t001:** Impact of different in situ synthesis conditions on AgNP content of fabric and washing durability of AgNP treatment.

Pretreatment	Synthesis Conditions	AgNP Content (µg g^−1^)	K/S of Sample	Gray Scale Ratings for Washing Durability after 20 Washes
None	Room temperature	51.69	1.04	2/3
	Hydrothermal	71.62	1.44	5
Mercerization	Room Temperature	77.42	1.56	3
	Hydrothermal	85.46	1.72	5

**Table 2 antibiotics-11-00864-t002:** UPF Values and Blocking for UVA (290–315 nm) and UVB (315–400 nm) for pristine cotton fabric and different samples of Cotton Fabrics with in situ AgNP synthesis (2 mM AgNO_3_, 20% Leaf *Azadirachta indica* leaf extract, Mixing Ratio 9:1, pH 7.1. A. No treatment B. No mercerization, Room temperature synthesis. C No mercerization, hydrothermal synthesis. D Mercerization, Room temperature synthesis. E Mercerization, Hydrothermal synthesis.

Sample	Ag Content (μg/g Fabric)	Transmission (UVA)	Transmission (UVB)	Blocking (UVA)	Blocking (UVB)	UPF
A	0	23.44	13.69	76.56	86.31	6.3
B	23.99	5.57	2.50	94.25	97.50	33.4
C	45.22	2.97	1.8	97.03	98.20	47.8
D	76.55	2.44	1.65	97.56	98.35	56.0
E	85.46	1.4	1.03	98.6	98.97	89.9

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
