# Peer review of "Development of Wash-Durable Antimicrobial Cotton Fabrics by In Situ Green Synthesis of Silver Nanoparticles and Investigation of Their Antimicrobial Efficacy against Drug-Resistant Bacteria"

_antibiotics, 2022, doi:10.3390/antibiotics11070864_

Round 1

Reviewer 1 Report

The authors of the manuscript titled "Development of Wash-durable Antimicrobial Cotton Fabrics by in-situ Green Synthesis of Silver Nanoparticles and Investigation of Their Antimicrobial Efficacy against Drug-Resistant Bacteria" propose a one-step preparation method to produce cotton fabric wih antibacterial properties using Azadirachta indica leaf extract as a reducing agent. The results are quite clearly presented and mostly accurately commented on, there are only several minor issues to be addressed in the manuscript:
It would be beneficial if a little more detailed characterisation of the form of the Ag nanoparticles was carried out. The size and shape of the nanoparticles have significant impact on their antibacterial activity, so these properties play an important role in the analysis of the antibacterial activity of the proposed material.

It should be described in a greater detail, how exactly the EDAX measurement data were used to evaluate the biofilm creation on the samples. Moreover, Fig. 8 containing the EDAX results should be reworked, the EDAX spectra are unreadable (cut the x-axis, enlarge the spectra) and the SEM images are too small.

Some of the presented data could be presented better - Table 1 and 2 should be merged, this would add just 1 column to the Table. The header of the table should cite just: "synthesis condition" and not the full experimental details on the rectant ratios, this will improve the formatting of the table. The statistical analysis description should be omitted from captions of Fig. 5 and 7, just the sentence: "Bars having different alphabet 290 letters are statistically different." should remain. There are too many arrows in Fig. 1B.

The conclusion states that the materials were effective in inhibiting the bacteria, however, the results show inhibition of only one of the bacterial strains. The conclusion should therefore state, that the prepared material could be effective against some antibiotic resistant bacteria.

Line 98 states that "leaf nanoparticles" were prepared by ball-milling. This expression is inaccurate, please revise.

There are some more minor errors in the manuscript: lines 159-165 are in a different typeface than the rest of the manuscript, line 429 should be 2.5 M NaOH, not 2.5 N NaOH, the correct title of the bacterial strain is S. epidermidis, not S. epidermis, header number 4.2.4.1 should be probably changed to 4.2.5, there are some typos in the Fig. 5 caption: thee -> three or the, expermiment -> experiment, there is probably a typo in line 48 cell well synthesis -> cell wall function.

Reviewer 2 Report

This manuscript investigated the “Development of Wash-durable antimicrobial cotton fabrics ……against drug resistant bacteria”. The subject of this paper is interesting and the results seem to support the enhancement of antibacterial activities. However, some data seem to be required more explanation to conclude their effects. Furthermore, I think some of the explanations are not clear for understanding. Thus, this manuscript needs major revision. The list of commends are shown below.

1.     In the 2nd paragraph of section 2.1, the experimental part will move to the experimental discussion.

2.     In 4th paragraph of section 2.1, why author give a general introduction about Mercerization in the results and discussion part? Better to move in the introduction section and discuss in more detail with recent references. Also, why the font style is different in this paragraph?

3.     The word “K/S value for unmercerized fabric was 1.04” is a repeating number of places.

4.     Line 174-176, how those refs [33,72] durability support your work? This is not clear.

5.     Author should provide a similar magnification range (Fig. 1) of with and without Ag treated samples for a fair comparison.

6.     The author mentioned “thick and uniform deposition”. However, the SEM figure looks at defects and cracks on the surface. Better to provide low and high magnified images and add clear results and discussion. What is the coating thickness of the sample? Add the details.

7.     Regarding AgNP deposition mechanism, The author mentioned “silver salt get deposited on cotton fabrics…….. nanoparticles that get instantly immobilized…. The mechanism is not clear. Is silver salt deposited first and then forming nanoparticles?

8.     What is the surface charge of present work samples? How about wet and dry conditions? Add results and discuss.

9.     Line 217-219, How the decreasing crystallinity can increase the availability of OH- groups? The evidence is not clear.

10.  Line 221, the author discusses increasing pore size, but the data is missing? How much of pore size increased?

11.  What is the swelling behavior of AgNP coated sample? Add the data and discuss. This will also be important for the coating stability and durability of the sample.

12.  Typo errors for the entire manuscript. For ex. (μg/g, OF SAMPLE, fabrics (, thesilver, etc.

13.  Line 227, “thereby ruling out any chemical bonding between fabric and AgNPs”. But the previous paragraph, the author mentioned “weak hydrogen bonds and Van der Waals force of attraction [61]”. The discussion is not clear, controversial and confusing.  

14.  Based on Fig. 3, the author should also provide EDX data to support the SEM image of “negligible loss” confirmation.

15.  Line 330-331, 339-340, cite the reference.

16.  Add EDX values of Fig. 8. The picture quality and its font is not clear. Need to replace. Moreover, the mechanism of antibacterial effects with biofilm resistance using Ag coating role on the fabrics is not clear and missing in the discussion.

17.  Rewrite the conclusion.

18.  Extensive editing of English language and style required. Thereby, the language should be smoothed so that this article can be more readable and understandable.

Round 2

Reviewer 2 Report

The author carefully revised and improved the quality of the manuscript. In my opinion, this paper should be suitable for publication in “Antibiotics”.